# The effect of Iran's health transformation plan on hospital performance: Kerman province

**Reza Goudarzi[1], Mohammad Tasavon Gholamhoseini[1]\*, Somayeh Noori Hekmat[2], Setareh YousefZadeh[3], Saeed Amini[4]**

**1** Health Services Management Research Center, Institute for Futures Studies in Health, Kerman University of Medical Sciences, Kerman, Iran, **2** Somayeh Noori Hekmat, Management and Leadership in Medical Education Research Center, Kerman University of Medical Sciences, Kerman, Iran, **3** Setareh YousefZadeh, Medical Informatics Research Center, Institute for Futures Studies in Health, Kerman University of Medical Sciences, Kerman, Iran, **4** Faculty of Health, Saeed Amini, Health Services Management Department, Arak University of Medical Sciences, Arak, Iran

\* mohamad.gholamhoseini@yahoo.com

**Data Availability Statement:** All relevant data are within the paper and at:(https://figshare.com/articles/dataset/dataset_xlsx/13550027/1).

## Abstract

Iran has performed Health Transformation Plan (HTP) from 2014 to obtain its defined goals. This study assesses and compares university and non-university hospitals' efficiency and productivity in Kerman provinces, Iran. The data of 19 selected hospitals, two years before and two years after Health Transformation Plan, was collected in this cross-sectional study. These data included the variables of physician and nurse number, and active beds as inputs and bed occupancy rate and inpatient admission adjusted with the length of stay as outputs. Data Envelopment Analysis method used to measure hospital efficiency. Malmquist Productivity Index is used to measure the efficiency change model before and after the plan. The efficiency and effect of the plan on hospitals' efficiency and productivity were assessed using R software. The results indicated that all hospitals' average efficiency before the HTP was 0.843 and after the HTP was increased to 0.874. However, it was not significant (P>0.05). Productivity also had a decreasing trend. Based on the DEA method results, it was found that university and non-university hospitals' efficiency and productivity did not increase significantly after the HTP. Therefore, it is recommended that attention be paid to hospitals' performance indicators regarding how resources are allocated and decisions made.

## Introduction

The increasing healthcare costs have persuaded the governments and health policymakers to increase productivity and efficiency [1]. Health system reforms more or less have left behind favorable effects. For example, after performing a health transformation program, Turkey's country has gained significant improvements regarding performance indices [2].

By attention to overall missions and upstream documents, especially Iran's 20 years' vision document and Iran supreme leader policies, the Iran Ministry of Health and Medical Education launched Health Transformation Plan(HTP) in 2014 [3]. The main goals of HTP included

**Funding:** The authors received no specific funding for this work.

**Competing interests:** The authors have declared that no competing interests exist.

**Abbreviations:** HTP, Health Transformation Plan; DEA, Data Envelopment Analysis; MPI, Malmquist Productivity Index; DMU, Decision-Making Unit; RTS, Return to Scale; CRS, Constant Return to Scale; VRS, Variable Return to Scale.

obtaining equity, financial protection, and access to healthcare services through performing 7 packages of decreasing payment rate of hospitalized patients, support physician residency in deprived areas, presence of resident specialist physicians in university hospitals, improving visit services quality, financial protection for difficult to cure patients, promotion hoteling services in the university hospitals and promote natural delivery [4].

Hospitals due to consuming the most resources in healthcare system, something from 50 to 80%, so promoting its efficiency is amongst the main goals of health policymakers worldwide. The conservative estimates indicate that about 300 billion dollars are annually missed because of inefficiency in hospital utilization [5,6].

On the one hand, health care managers need to make decisions to identify problems at the first stage. They should decide to design the solutions; finally, they should decide to present the appropriate responses. On the other hand, they need to make optimal resource allocation decisions, identify efficient and non-efficient units, their strengths and weaknesses; finally, the correct formulation of health system strategies [7]. Beyond all these issues, the managers have limited time and capacity which should make decisions in a limited time frame with the highest quality and present decisions in a plausible way. To solve the problems, some methods and software were designed to help managers in decision-making. As the output of these methods is obtained without men's intervention, so they are accepted by all healthcare managers and staff [8,9].

A Decision-Making Unit (DMU) such as a hospital is efficient when a predefined level of its outputs is produced with the lowest inputs [10]. In this regard, there are different methods to assess hospital efficiency, including Data Envelopment Analysis (DEA) [11]. As a non-parametric linear programming method, DEA has unique measures such as simultaneous analysis of several inputs and outputs that differentiate it from other efficiency measuring methods. By attention to Return to Scale (RTS), DEA includes two models; Constant Return to Scale (CRS), which is suitable when all DMUs work at the optimum level, and Variable Return to Scale (VRS), which is suitable when all DMUs do not work in optimum level [7,12].

The study of Pirani et al. in southwest of Iran in 2018 [13], the study of Moradi et al. in Kurdistan of Iran in 2017 [14], the study of Samut and Cafrı in OECD countries in 2016 [15], the study of Van Ineveld et al. in the Netherlands in 2016 [16], the study of Sahin Gok and Altındag in Turkey in 2015 [1], the study of Azar et al. in Tehran in 2013 [17] have assessed the effect of HTP on hospital performance using DEA method. Li et al. in Shandong Province were also evaluated the efficacy of county public hospitals following China's new medical reform [18]. Another study in China assessed health system productivity pre-and post-2009 healthcare reform [19].

In Iran, few studies have examined the impact of HTP on hospitals' efficiency and productivity. Thus, this paper aims to compare the efficiency of university and non-university hospitals in Kerman before and after HTP.

## Materials and methods

### Study population and sampling

The study population of this cross-sectional study includes 24 hospitals located in Kerman province in southwestern Iran. Five hospitals were excluded from the study due to insufficient data in Kerman University of Medical Sciences databases. So, we do not use sampling, and all 19 hospitals were included in the study.

Ten hospitals were university, and nine were non-university (public and private). Ethics Committee of Kerman University of Medical Sciences approved this study on the 8th of December in 2019 (No. IR.KMU.REC.1398.431).

## DEA method

The non-parametric method of DEA is used to measure efficiency and productivity. In this method, it is possible to determine efficient points using two hypotheses of CRS and VRS, and to determine the efficiency DMUs, it is possible to use two hypotheses input-oriented minimization and output-oriented maximization [20,21].

Because 1 unit increases in the inputs, the outputs do not increase the same, so the VRS method is used to assess efficiency. Also, because the outputs are not in managers' control, they can increase efficiency only by minimizing the inputs-oriented model used to analyze using the DEA program [7]. The input-oriented linear programming of VRS model is shown below:

$$Min_{\lambda,OS,IS} \left( M'_1.OS + K'_1.IS \right)$$
$$st : -y_i + Y\lambda - OS = 0,$$
$$\theta x_i - X\lambda - OS = 0$$
$$N'_1.\lambda \leq 0, \lambda \geq 0, OS \geq 0, IS \geq 0$$

(1)

Where $\theta$ is a scalar, $\lambda$ is a N×1 vector of consonants and y represents the output vector which can be produced using input vector x. OS is an M×1 vector of output slacks, IS is a K×1 vector of input slacks, and M1 and K1 are M×1 and K×1 vectors of ones, respectively.

Another measure used in this study is Malmquist Index (MI), which evaluates the efficiency changes over time [22]. MI separates total productivity into two main ingredients. Technological efficiency changes and technical efficiency changes. On the one hand, if MI due to the input-oriented method is lower than one, it implies performance improvement. While, the MI higher than one implies a decrease in performance over time. On the other hand, due to output-oriented method, MI lower than one implies worsening performance, and bigger than one indicated improvement in performance over time [23,24]. MI was used in the current study to assess changes in hospital efficiency before and after HTP.

## Data source

The most important inputs and outputs to assess hospital performance were identified by a literature review [1,21,25,26]. Then, the data regarding selected parameters in the study hospitals were extracted from Kerman University of Medical Sciences databases for a period of two years before and two years after HTP in 2014.

## Model inputs and outputs

To assess hospital performance using the DEA method, the indices were categorized into inputs and outputs. Input variables included the number of physicians, nurses, active beds, and outputs variables included bed occupancy rate and inpatient admission. It is worth noting that the admission variable was adjusted due to length of stay in hospital.

## Data analysis

After performing Kolmogorov–Smirnov test to assess normality, the normal data using Paired t-test and otherwise, the data were analyzed using the Wilcoxon test to compare mean efficiency and productivity of hospitals in two mentioned periods. In this study, the efficiency data of the first scenario (university hospitals) had a normal distribution, so we used the paired t-test to measure changes in efficiency before and after HTP. However, the efficiency data of second and third scenarios (non-university hospitals and all hospitals) were abnormally

distributed, so we used Wilcoxon test to measure changes in efficiency before and after HTP. R software was used to calculate the efficiency and productivity of hospitals.

## Results

Nineteen hospitals in which 10 were university and nine non-university hospitals were assessed. Before HTP, 70% and 78% of university and non-university hospitals obtained optimum efficiency scores, respectively (score between 0.8 and 1). This score after HTP for university and non-university hospitals was 80% and 78%, respectively.

Table 1 indicates that the inputs and output are compared in 3 scenarios, including university hospitals, non-university hospitals, and total university hospitals before and after HTP. The results of paired t-test and Wilcoxon test showed that HTP has significantly increased the inputs of nurses and active beds and the inputs of bed occupancy rate and the number of inpatient admissions adjusted with stay length in university hospitals (P<0.05). Also, bed occupancy rate and the number of inpatient admissions adjusted with stay length have increased by 15% and 20% after HTP, respectively among the positive effects of HTP.

There was no significant increase in the inputs and outputs after HTP in non-university hospitals (P<0.05) which means that HTP has not caused a significant change in the inputs and outputs of non-university hospitals.

The result of HTP effect on total Kerman province hospitals indicated that the most change was in inpatient admission adjusted with stay length with 20% increase and the lowest change was in the number of physicians with 0.4% decrease. An increase in the inputs of nurses and active beds and inpatient admission adjusted with stay length was statistically significant (P<0.05).

**Table 1. Comparing the mean of inputs and outputs of university and non-university hospitals before and after HTP.**

| Inputs | Mean before HTP | Mean after HTP | Tests (sig) |
|---|---|---|---|
| **University hospitals** | | | |
| Physician | 12.40 | 11.80 | Paired t-test (0.749) |
| Nurse | 141.85 | 174.60 | Wilcoxon (0.013) |
| Active beds | 179.30 | 198 | Wilcoxon (0.005) |
| Outputs | | | |
| Bed occupancy rate | 56.31 | 65.03 | Paired t-test (0.015) |
| Inpatient admission adjusted with stay length | 41022.65 | 49287.98 | Wilcoxon (0.005) |
| **Non-university hospitals** | | | |
| Physician | 9.17 | 8.89 | Wilcoxon (0.715) |
| Nurse | 74.33 | 91.44 | Paired t-test (0.184) |
| Active beds | 85.94 | 102.89 | Wilcoxon (0.528) |
| Outputs | | | |
| Bed occupancy rate | 49.14 | 50.56 | Paired t-test (0.765) |
| Inpatient admission adjusted with stay length | 15700.01 | 19236.26 | Wilcoxon (0.139) |
| **Total hospitals** | | | |
| Physician | 10.87 | 10.42 | Wilcoxon (0.624) |
| Nurse | 109.87 | 135.21 | Wilcoxon (0.003) |
| Active beds | 135.08 | 152.95 | Wilcoxon (0.008) |
| Outputs | | | |
| Bed occupancy rate | 52.92 | 58.18 | Paired t-test (0.069) |
| Inpatient admission adjusted with stay length | 29027.71 | 35052.96 | Wilcoxon (0.001) |

**Table 2. Technical efficiency of university, non-university and total hospitals.**

| Study scenarios | Before HTP | | | After HTP | | | Test (sig) |
|---|---|---|---|---|---|---|---|
| | 2012 | 2013 | Mean | 2015 | 2016 | Mean | |
| **University hospitals** | 0.861 | 0.828 | 0.845 | 0.877 | 0.858 | 0.868 | Paired t test (0.548) |
| **Non-university hospitals** | 0.835 | 0.849 | 0.842 | 0.875 | 0.886 | 0.880 | Wilcoxon (0.686) |
| **Total hospitals** | 0.849 | 0.838 | 0.843 | 0.876 | 0.871 | 0.874 | Wilcoxon (0.294) |

In general, the results indicated that non-university hospitals had obtained higher efficiency after HTP compared to other hospitals (Table 2). The difference between the mean efficiency score of studied hospitals (university, non-university, and total) before and after HTP was not statistically significant (p<0.05), which means that increase in the average efficiency of hospitals is due to random effect and reasons other than HTP can lead to this increase. The mean efficiency score of university hospitals in years after HTP id est. 2015 and 2016 had decreased from 0.877 to 0.858, respectively. The efficiency of non-university hospitals slightly increased in 2016 compared to 2015. Also, considering total hospitals, efficiency score has increased 3% after HTP compared to before it (Fig 1). The mean efficiency score of university and non-university hospitals was 0.858 and 0.886 in 2016, respectively, indicating hospitals' efficiency promotion capacity without any decrease in the costs and applying the same amount of inputs was 14.2% and 11.4%, respectively (Table 2).

The productivity of the mentioned scenarios which were calculated two years before and two years after HTP are presented in Table 3. In all scenarios, hospitals in 2013 had low performance compared to 2012. The situation had become a little better in 2016, and hospital performance improved (especially in non-university hospitals), productivity has decreased. Mean productivity before and after HTP was significant only in the second scenario (non-university hospitals) (p = 0.046). It can be concluded that efficiency has increased after HTP, but productivity has a decreasing trend.

The average productivity in various scenarios is shown in Fig 2. According to the study assumption (input-oriented method), decreasing in productivity value means that productivity of DMUs has improved and therefore in all scenarios, productivity has improved after the HTP although it is not statistically significant in some cases.

## Discussion

This study indicated the comparison between the university and non-university Kerman hospitals' efficiency before and after HTP using the non-parametric method of DEA approach and the productivity of hospitals using the Malmquist Index between years 2012–2016. Health

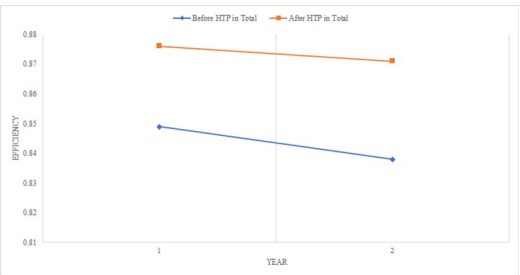

**Fig 1. Efficiency of total hospitals before and after HTP.**

**Table 3. The comparison of Malmquist productivity index of hospitals before and after HTP.**

| Study scenarios | Before HTP | | After HTP | | Test (sig) |
| --- | --- | --- | --- | --- | --- |
| | Mean | SE | Mean | SE | |
| **University hospitals** | 1.723 | 0.151 | 1.517 | 0.177 | Wilcoxon) 0.308( |
| **Non-university hospitals** | 1.711 | 0.256 | 1.187 | 0.057 | Paired t-test (0.046) |
| **Total hospitals** | 1.800 | 0.159 | 1.449 | 0.138 | Paired t-test (0.058) |

system reforms, including HTP encourage hospitals at the same time to have higher efficiency with higher quality in the services [11].

The results indicated that mean of inputs (excluding the number of physicians) and outputs after performing HTP have increased for university and non-university hospitals. These increases were significant in some cases. So, the number of hospital beds and admissions adjusted with mean stay length have increased after HTP significantly in university hospitals. In two separate studies, Piroozi et al. [27] and Beiranvand et al. [28] analyzed the HTP hospitalization rate effects in Iran and showed hospitalization rates increased after HTP significantly.

By attention to the observed significant difference between the inputs (the number of active beds) and the outputs (admission adjusted with a mean length of stay) in university hospitals after HTP, it can be concluded that since university hospitals are among university and great ones and also since there is the relationship between the efficiency and size of hospitals (for example hospitals with 200–400 beds have higher efficiency than hospitals above 400 and lower 200 beds) [29,30], so university hospitals after HTP have obtained higher accessibility to the inputs than other hospitals.

Performing HTP packages were accompanied by a decrease in patient payment and an increase in access to the services. These factors have increased patients' burden of visit to hospitals, long waiting lists, and as a result, an increase in inpatient admission adjusted with the mean length of stay in university hospitals than non-university ones. Furthermore, it can be said cautiously that HTP has had a significant effect on the inputs and outputs. The study of Sahin et al. showed that the number of nurses and the average number of inpatient and outpatients after the HTP increased significantly [31], which is in agreement with our findings.

Despite the increase in inputs and outputs, hospitals' increases after the HTP implementation period than before were not significant. The reason was due to small changes in the number of physicians. Therefore, the physician variable may be the most important variable influencing hospitals efficiency.

A research in Greece comparing the impact of pre-and post-health reform on 111 public hospitals in Greece in 2010 found that health reform has increased efficiency in the short term [32]. Kakemam and Darghahi also showed that the average efficiency of public hospitals in Iran has increased after HTP [33], while another study in Turkey after launching Turkish

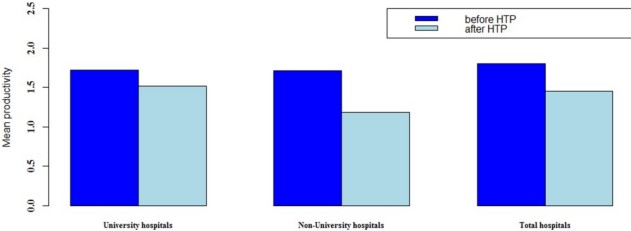

**Fig 2. Average productivity of different scenarios before and after HTP.**

health reforms indicated that the efficiency of university hospitals has increased and it has decreased for private hospitals [1]. Research in Japan has shown that the law-application system reforms are not enough to increase hospitals' efficiency and the researchers concluded that a systematic approach should be considered in order to improve efficiency.[34].

By attention to being the DEA method as input-oriented, an increase in productivity trend indicates a decline in hospital performance, which can be inferred that changes in productivity may be due to creating a shock or HTP effect on the inputs.

The trend of changes in DMUs efficiency using Malmquist Index was assessed due to 2012. The results indicated that productivity trend after HTP in hospital universities declined. In non-university hospitals, although performance slightly improved after HTP and its changes were statistically significant, productivity declined. It can be concluded that HTP had no major impact on trends in the productivity of hospitals. However, this impression should be cautiously reported due to the short study time. How to plan and implement health system reforms are among the determinants of reform outcomes.

Similar findings of reforms are not consistent to the current study. Studies in Turkey showed that HTP implementation in this country was successful and had a significant impact on hospitals' efficiency and productivity. Mollahaliloglu concluded that hospital efficiency and productivity increased following HTP implementation [35]. The Sahin study has shown that Turkish public hospitals' productivity improved from 2005 to 2008 [31]. Hospital productivity in Vietnam has also been shown to improve the following reforms due to structural changes in public hospitals [36]. One reason for disparity between the present study and other studies is that Iran's HTP has faced the challenge of physician shortage [37]. Thus, the productivity of hospitals has decreased.

Among different hospital efficiency assessment methods, the DEA method is the most beneficial one [38]. One of DEA's unique features than other methods is the simultaneous analysis of several inputs and several outputs [12] which determine efficiency as the ratio among corresponding weights of outputs to corresponding weights of inputs [39]. Also, through precise estimation of efficiency, it can provide each hospital's comparability with the peer ones [40]. Another advantage of the DEA method than others in efficiency analysis is the determination of surplus production factors in hospitals or other DMUs, as this can be used in other sectors such as banks and financial services, investment companies, and transport and shipping [41]. It is a managerial method which presents the solutions and is suitable for not-for-profit entities and hospitals whose services are not possible for precise pricing [42].

This study's strengths can point out simultaneous measurement and analysis of several inputs and outputs and precise calculation of efficiency and productivity in university and non-university hospitals before and after HTP. The absence of a case-mix index in the output of hospitals, the absence of permanent physicians as an essential hospital resource due to physician workflow in different hospitals, and lastly, not checking the impact of factors in the external environment on the efficiency of studies hospitals are among the weaknesses.

## Conclusion

In general, this study showed that the HTP had not had a significant impact on the university and non-university hospitals' efficiency, and the productivity of hospitals has not significantly improved. Support plans such as HTP may be encountered with a decrease in efficiency if hospitals cannot use the resources to provide higher quality services for more patients. So, it is proposed to allocate resources to the hospitals due to assessment performance indices in the previous periods and rooting the issues to obtain higher efficiency. Given that this study was conducted in one of the provinces of Iran, therefore, to generalize the results should be

considered cautiously. Hence, broader empowerment of local healthcare officials to make decisions regarding how to allocate the university resources by attention to the needs and necessities and then assessing changes in hospitals' efficiency score, providing feedback for hospital managers and supporting interventional plans to improve performance seems necessary.

## Author Contributions

**Conceptualization:** Reza Goudarzi, Mohammad Tasavon Gholamhoseini, Somayeh Noori Hekmat.

**Data curation:** Reza Goudarzi, Somayeh Noori Hekmat.

**Formal analysis:** Reza Goudarzi, Mohammad Tasavon Gholamhoseini.

**Investigation:** Somayeh Noori Hekmat.

**Methodology:** Mohammad Tasavon Gholamhoseini.

**Project administration:** Reza Goudarzi.

**Software:** Mohammad Tasavon Gholamhoseini.

**Writing – original draft:** Mohammad Tasavon Gholamhoseini, Setareh YousefZadeh.

**Writing – review & editing:** Mohammad Tasavon Gholamhoseini, Saeed Amini.

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
