## [Decision Letter · Decision Letter 0]

3 Sep 2020

PONE-D-20-18684

The effect of Iran's Health Transformation Plan on Hospital Performance Kerman Province

PLOS ONE

Dear Dr. Tasavon Gholamhoseini,

Thank you for submitting your manuscript to PLOS ONE. After careful consideration, we feel that it has merit but does not fully meet PLOS ONE’s publication criteria as it currently stands. Therefore, we invite you to submit a revised version of the manuscript that addresses the points raised during the review process.

We look forward to receiving your revised manuscript.

Kind regards,

Edris Hasanpoor

Academic Editor

PLOS ONE

Journal Requirements:

3. Your ethics statement must appear in the Methods section of your manuscript. If your ethics statement is written in any section besides the Methods, please move it to the Methods section and delete it from any other section. Please also ensure that your ethics statement is included in your manuscript, as the ethics section of your online submission will not be published alongside your manuscript.

Reviewers' comments:

Reviewer's Responses to Questions

**Comments to the Author**

1. Is the manuscript technically sound, and do the data support the conclusions?

Reviewer #1: Partly

Reviewer #2: Partly

2. Has the statistical analysis been performed appropriately and rigorously? 

Reviewer #1: No

Reviewer #2: Yes

3. Have the authors made all data underlying the findings in their manuscript fully available?

Reviewer #1: Yes

Reviewer #2: Yes

4. Is the manuscript presented in an intelligible fashion and written in standard English?

Reviewer #1: Yes

Reviewer #2: No

5. Review Comments to the Author

Reviewer #1: In the methods section, the authors mentioned that 19 selected hospitals are included in the study. What is the total number of the hospitals in the province? Is the 19 hospitals representative number or not?

Results: There is no explanation why 1) the average efficiency was not significant. Can you explain this result in terms of the results of the table 1 which includes significant tests with p-value <0.05.

Table 2 is not very clear. It does not include any significant results, that is technical efficiency has no significant change. Can you connect table 2 results with table 3? Is there statistical significant change of the Malmquist productivity index between years?

In table 2, university hospitals efficiency is done with Paired t -test. The non university hospitals test is wilcoxon. When you test total hospitals you use Wilcoxon. Don't you think this is inconsistency? in the first case you have normality assumption and the other 2 categories you use non parametric test. Please explain!!!

Reviewer #2: 1. The introduction section needs to be clarified for the readers for better understanding. There is need of more references so far decision making system is concerned. Literature are available in the context of global level in decision making.

2. In methodology the authors should explain the procedures in detail so that the study can be replicated elsewhere. for example DEA method needs more specification. Data analyse procedures also need details explanation.

3. In result table 1 is understandable But, table 2 and 3 need explanation and scientific interpretation of the results. there is also need of clarity do the understanding of the readers.

4. the discussion section needs complete overwhelming. Similar results are to be interpreted with study of more literature. Important results from the tables are still missing in discussion section

6. PLOS authors have the option to publish the peer review history of their article (what does this mean?). If published, this will include your full peer review and any attached files.

Reviewer #1: No

Reviewer #2: **Yes: **Ranjit Kumar Dehury

---

## [Author Response · Author response to Decision Letter 0]

22 Oct 2020

Reviewer#1

1. In the methods section, the authors mentioned that 19 selected hospitals are included in the study. What is the total number of the hospitals in the province? Is the 19 hospitals representative number or not?

Response: Thanks for your comment. In our revised version, we provide more detail about hospital data. A total of 24 hospitals were entered to study, 5 of which were excluded from the final analysis due to lack of sufficient data (Please see page 5, lines 86-89).

2. There is no explanation why 1) the average efficiency was not significant. Can you explain this result in terms of the results of the table 1 which includes significant tests with p-value <0.05.

Response: Thanks for your comment. Although some variables changed significantly after HTP, the physician variable was not significant and thus the efficiency and productivity changes after HTP were not significant. We discuss this in the Discussion section of the revised manuscript (Please see pages 12 and 13, lines 216 and 240-242).

 We also performed a regression to find that the physician variable could be more important than the other variables( Please see the figure below).

3. Table 2 is not very clear. It does not include any significant results, that is technical efficiency has no significant change. Can you connect table 2 results with table 3? Is there statistical significant change of the Malmquist productivity index between years?

Response: Thanks for your comment. We have changed the Table 2 as suggested by reviewer #1.

Statistical tests were performed for productivity changes before and after HTP and were reported in Table 3.

4. In table 2, university hospitals efficiency is done with Paired t -test. The non university hospitals test is wilcoxon. When you test total hospitals you use Wilcoxon. Don't you think this is inconsistency? in the first case you have normality assumption and the other 2 categories you use non parametric test. Please explain!!! 

Response: Thanks for your comment. We first assessed the normality and abnormality of the statistical distribution of data in each scenario using the Kolmogorov – Smirnov test; then, we measured the efficiency and productivity changes by appropriate statistical tests ( Paired t-test or Wilcoxon). In this study, the efficiency data of university hospitals had a normal distribution, so we used the paired t-test to measure changes in efficiency before and after HTP.

However, the efficiency data of non-university hospitals were abnormally distributed, and the Wilcoxon test was used. The efficiency data of all hospitals were also abnormally distributed. We have explained more in the revised version (Please see page 7, lines 130-136).

Reviewer#2

1. The introduction section needs to be clarified for the readers for better understanding. There is need of more references so far decision making system is concerned. Literature are available in the context of global level in decision making.

Response: Thanks for your comment. As suggested, we have added details on the importance of decision-making in the health system (Please see pages 3 and 4, lines 59-67).

2. In methodology the authors should explain the procedures in detail so that the study can be replicated elsewhere. for example DEA method needs more specification. Data analyse procedures also need details explanation.

Response: Thanks for your comment. We have added more details in the revised version (Please see pages 5-7, lines 101-109 and 132-136).

3. In result table 1 is understandable But, table 2 and 3 need explanation and scientific interpretation of the results. there is also need of clarity do the understanding of the readers.

Response: Thanks for your comment. We have changed Tables 2 and 3 as suggested by reviewer #2.

4. the discussion section needs complete overwhelming. Similar results are to be interpreted with study of more literature. Important results from the tables are still missing in discussion section

Response: Thanks for your comment. As suggested, the discussion section is updated in the revised version of the paper.

---

## [Decision Letter · Decision Letter 1]

23 Dec 2020

PONE-D-20-18684R1

The effect of Iran's Health Transformation Plan on Hospital Performance Kerman Province

PLOS ONE

Dear Dr. Mohamad Tasavon Gholamhoseini,

Thank you for submitting your manuscript to PLOS ONE. After careful consideration, we feel that it has merit but does not fully meet PLOS ONE’s publication criteria as it currently stands. Therefore, we invite you to submit a revised version of the manuscript that addresses the points raised during the review process.

We look forward to receiving your revised manuscript.

Kind regards,

Sharon Mary Brownie

Academic Editor

PLOS ONE

Reviewers' comments:

Reviewer's Responses to Questions

**Comments to the Author**

1. If the authors have adequately addressed your comments raised in a previous round of review and you feel that this manuscript is now acceptable for publication, you may indicate that here to bypass the “Comments to the Author” section, enter your conflict of interest statement in the “Confidential to Editor” section, and submit your "Accept" recommendation.

Reviewer #1: All comments have been addressed

Reviewer #2: All comments have been addressed

2. Is the manuscript technically sound, and do the data support the conclusions?

Reviewer #1: Yes

Reviewer #2: Partly

3. Has the statistical analysis been performed appropriately and rigorously? 

Reviewer #1: I Don't Know

Reviewer #2: Yes

4. Have the authors made all data underlying the findings in their manuscript fully available?

Reviewer #1: Yes

Reviewer #2: Yes

5. Is the manuscript presented in an intelligible fashion and written in standard English?

Reviewer #1: Yes

Reviewer #2: Yes

6. Review Comments to the Author

Reviewer #1: I think if you use graphical representation for the productivity (eg histograms or boxplots) before and after HTP, the readers would get better understanding of your variables.

Can you report the standard error of the productivity numbers in table 3? This could tell us important information about the variability of productivity for the suggested 4 categories

Reviewer #2: 1. The article need more discussion to make the study comparable with others.

2. the reference articles have to be from very high authoritative sources. New articles should be added in introduction section to make the technicalities understandable.

3. The tables especially paired test tables have to be explained for understanding.

7. PLOS authors have the option to publish the peer review history of their article (what does this mean?). If published, this will include your full peer review and any attached files.

Reviewer #1: No

Reviewer #2: **Yes: **Ranjit Kumar Dehury

---

## [Author Response · Author response to Decision Letter 1]

13 Jan 2021

Dear editor in chief

Thank you for your invaluable comments and guidance in improving this manuscript. It should be noted that based on the Reviewer’s comments and according to Journal requirements, we made some changes in manuscripts that are specified in text. We presented the responses to reviewers’ comments separately in the following table. I as the corresponding author on behalf of all authors express our readiness to do any further revision seems necessary by the journal.

Reviewer#1-1 : I think if you use graphical representation for the productivity (eg histograms or boxplots) before and after HTP, the readers would get better understanding of your variables.

Response: Thanks for your comment. We added two figures in our revised version so that the reader can understand the changes in productivity and efficiency better (please see figures 1 and 2 and see page 11, lines 193-196).

Reviewer #1-2: Can you report the standard error of the productivity numbers in table 3? This could tell us important information about the variability of productivity for the suggested 4 categories.

Response : Thanks for your comment. We added standard errors of productivity numbers for different scenarios (please see table 3).

Reviewer #2-1: The article need more discussion to make the study comparable with others.

Response: Thanks for your comment. We added some studies to compare with the present study in the discussion section (please see page 12, lines 208-210 and page 13, lines 234-236).

Reviewer #2-2: 2. the reference articles have to be from very high authoritative sources. New articles should be added in introduction section to make the technicalities understandable.

Response: Thanks for your comment. We searched Scopus and Pubmed databases for the most recent studies related to this research, and two studies were added to the introduction section (please see page 4, lines 80-82).

Reviewer #2-3: 3. The tables especially paired test tables have to be explained for understanding.

Response: Thanks for your comment. We provided more details about statistical tests in our revised version (please see page 9, lines 181, 196, and page 10, line 197).

---

## [Decision Letter · Decision Letter 2]

3 Feb 2021

The effect of Iran's Health Transformation Plan on Hospital Performance Kerman Province

PONE-D-20-18684R2

Dear Dr. Mohamad Tasavon Gholamhoseini,

We’re pleased to inform you that your manuscript has been judged scientifically suitable for publication and will be formally accepted for publication once it meets all outstanding technical requirements.

Kind regards,

Sharon Mary Brownie

Academic Editor

PLOS ONE

Reviewers' comments:

Reviewer's Responses to Questions

**Comments to the Author**

1. If the authors have adequately addressed your comments raised in a previous round of review and you feel that this manuscript is now acceptable for publication, you may indicate that here to bypass the “Comments to the Author” section, enter your conflict of interest statement in the “Confidential to Editor” section, and submit your "Accept" recommendation.

Reviewer #1: All comments have been addressed

Reviewer #2: All comments have been addressed

2. Is the manuscript technically sound, and do the data support the conclusions?

Reviewer #1: Partly

Reviewer #2: Yes

3. Has the statistical analysis been performed appropriately and rigorously? 

Reviewer #1: I Don't Know

Reviewer #2: Yes

4. Have the authors made all data underlying the findings in their manuscript fully available?

Reviewer #1: Yes

Reviewer #2: Yes

5. Is the manuscript presented in an intelligible fashion and written in standard English?

Reviewer #1: Yes

Reviewer #2: Yes

6. Review Comments to the Author

Reviewer #1: Authors have addressed all the issues but they have not discussed the additional information (standard errors and the graphics)

Reviewer #2: Comments have been addressed. But, more references can be discussed for improvement of the article. The language of the article can also be improved for the readers. The technical words should also be explained for betterment.

7. PLOS authors have the option to publish the peer review history of their article (what does this mean?). If published, this will include your full peer review and any attached files.

Reviewer #1: No

Reviewer #2: **Yes: **RANJIT DEHURY

---

## [Editor Report · Acceptance letter]

5 Feb 2021

PONE-D-20-18684R2 

The effect of Iran's Health Transformation Plan on Hospital Performance:
Kerman Province 

Dear Dr. Tasavon Gholamhoseini:

I'm pleased to inform you that your manuscript has been deemed suitable for publication in PLOS ONE. Congratulations! Your manuscript is now with our production department. 

Kind regards, 

on behalf of

Professor Sharon Mary Brownie 

Academic Editor

PLOS ONE